# Differential Impact of Risk Factors for Cognitive Decline in Heterosexual and Sexual Minority Older Adults in England

**DOI:** 10.3390/brainsci15010090

**Published:** 2025-01-18

**Authors:** Riccardo Manca, Jason D. Flatt, Annalena Venneri

**Affiliations:** 1Department of Life Sciences, Brunel University of London, Uxbridge UB8 3PH, UK; riccardo.manca@brunel.ac.uk; 2Department of Medicine and Surgery, University of Parma, 43125 Parma, Italy; 3School of Public Health, University of Nevada Las Vegas, Las Vegas, NV 89119, USA; jason.flatt@unlv.edu

**Keywords:** cognitive decline, memory, sexual orientation, minority, mental health, marital status, loneliness, socio-economic status

## Abstract

Background/Objectives: Sexual minority older adults (SMOAs) report greater subjective cognitive decline (SCD) than heterosexual older adults (HOAs). This study aimed to compare the impact of multiple psycho-social risk factors on objective and subjective cognitive decline in HOAs and SMOAs. Methods: Two samples of self-identified HOAs and SMOAs were selected from the English Longitudinal Study of Ageing. Reliable change indices for episodic and semantic memory were created to assess cognitive decline. SCD was self-reported for memory and general cognition. Depressive symptoms, loneliness, marital status and socio-economic status were investigated as risk factors. Results: No between-group differences were found in cognitive decline. Higher depression was associated with greater SCD risk and worse semantic memory decline. The latter effect was stronger in SMOAs. The findings were largely replicated in the sensitivity analysis. Conclusions: Poor mental health may represent the strongest driver of cognitive decline in SMOAs and to a greater extent than in HOAs.

## 1. Introduction

Advancing researchers’ and healthcare professionals’ understanding of specific determinants of health in diverse populations is a crucial step to achieve equity in diagnosis, prevention and treatment [1]. However, sexual and gender minority (SGM) older adults (i.e., lesbian, gay, bisexual, transgender, queer, intersex, asexual and additional identities people) remain under-represented in neurological research. Investigations on cognitive decline and dementia risk are lacking in this population compared with studies on the consequences of neuro-infectious diseases, especially of human immunodeficiency virus [2].

Research on the cognitive health of SGM older adults has only developed in the past few years, and the currently available publications have several methodological limitations [3]. Thus, many questions on possible inequalities and the factors underlying them remain currently unaddressed [4]. For sexual minority older adults (SMOAs), in particular, inconsistent findings have emerged on differences in the rates of both subjective (SCD) and objectively assessed cognitive decline compared with heterosexual older adults (HOAs).

Most studies found no differences in SCD rates between SMOAs and HOAs [5,6,7]. Even methodologically similar investigations based on data from the same database (i.e., the Behavioral Risk Factor Surveillance System) have found inconsistent results, either reporting higher SCD rates in SMOAs than in HOAs [8] or no between-group differences [5]. It is possible that the SCD risk may be higher in specific sub-groups, such as SM women [9] and, more specifically, among bisexual compared with lesbian older women [10]. The literature, therefore, provides limited insights into the risk of dementia in SMOAs, since the subjective perception of cognitive decline can be influenced by several factors, including depression [11], childhood sexual abuse [12] and a lack of co-worker support of SGM identity [13]. One study, however, found that higher psychological distress did not fully explain greater SCD risk in SMOAs [14]. Specific risk/protective factors for SCD have not always been addressed [6,7], and none of the studies currently available included either indices of objective cognitive performance or biomarkers of neurodegeneration.

For objective cognitive health, instead, previous cross-sectional investigations of performance on neuropsychological tests found either no differences [15] or better episodic memory in SMOAs than in HOAs [16,17,18]. A handful of longitudinal studies on cognitive decline in SMOAs focussed on people either in same-sex (SSRs) or different-sex relationships (DSRs). Perales-Puchault et al. [19] observed no differences in the risk of either mild cognitive decline (MCI) or dementia, while a more recent study found milder decline in attention/working memory for people with dementia in SSRs compared with those in DSRs [20]. Hanes and Clouston [21] have also shown that older adults in SSRs had slower rates of cognitive decline but had worse cognitive profiles at baseline. Moreover, people in SSRs included in this study met the criteria for both MCI and possible dementia at a younger age than people in DSRs. The only study that investigated self-reported sexual orientation as one of many potential factors associated with cognitive decline in older adults found no significant differences in changes in episodic memory between SMOAs and HOAs [18].

A UK-based primary care survey may offer some insights into why no objective evidence of cognitive health disparities have been found so far between SMOAs and HOAs: Saunders et al. [22] found a higher dementia risk in non-heterosexual than in heterosexual people younger than 55 years of age, but not among older adults. This finding suggests that risk factors other than neurodegeneration may increase the risk of cognitive decline in non-heterosexual people. Moreover, both Saunders et al. [22] and a more recent USA-based study found that dementia risk is increased among lesbian/bisexual women but not among non-heterosexual men [23]. However, the exploration of risk factors in SMOAs has been overlooked by most longitudinal studies [18,19,20]. Hanes and Clouston [21] observed that differences in cognitive decline between older adults in SSRs and DSRs were significant only among participants with an educational attainment below university degree and with a higher household income. A couple of cross-sectional studies have shown that worse mental and physical health, being single/not married and a lack of social connections may be associated with higher odds of objective cognitive impairment in SMOAs [24,25].

The predominant theorical framework that has been proposed to explain health inequalities in the SGM population is minority stress, i.e., chronic stress associated with the stigma experienced by people due to their minority status [26]. According to this framework, most people from minorities experience both distal (e.g., discrimination and microaggressions) and proximal stressors (e.g., internalized homophobia and hypervigilance) in their social environments. All these factors can have negative effects on the mental and physical health of SGM populations. Although minority stress was suggested as a contributor to a greater risk of cognitive decline in SGM older adults [27], only one cross-sectional study has investigated and found a significant, yet small, negative effect of discrimination related to sexual orientation on fluid intelligence of SMOAs [16]. The scarcity of research on minority stress and cognitive health depends primarily on the fact that all studies in this field have used public datasets that lacked SMOA-relevant variables.

The abovementioned studies provide some limited sources of evidence that suggest how different social, relational and psychological factors may influence cognitive health in SMOAs. However, most risk factors have been only investigated in single studies, and replication in independent cohorts is needed, considering the lack of sample diversity observed [3]. Most risk factors investigated previously, although they may represent greater issues for SMOAs (e.g., worse mental health [28] and a higher likelihood of being single [29]), are not SMOA-specific. However, minority stressors may exert an effect on multiple psycho-social risk factors with potential differential effects between SMOAs and HOAs that have not been addressed yet. Finally, the quantification of the relative strength of the impact of such psycho-social risk factors on either the subjective or objective cognitive health of SMOAs has yet to be elucidated. For these reasons, to date, no definite conclusions can be drawn on (1) whether SMOAs either experience greater changes in objective measures of cognitive performance or are only more likely to report SCD than HOAs and, if there are differences, (2) what factors may explain such disparities.

Since psycho-social risks are, in most cases, modifiable, elucidating what factors may be affecting the cognitive health of SMOAs more strongly is of primary importance to design targeted programmes to prevent decline. Therefore, this study aimed to compare the impact of multiple psycho-social risk factors (either subjectively reported or objectively assessed) between cognitively healthy SMOAs and HOAs, on:Two measures of SCD for memory and for global cognition (Aim 1);Two objective measures of cognitive decline in episodic and semantic memory (Aim 2).

## 2. Materials and Methods

### 2.1. Participant Sample

The sample used in this study was selected from the English Longitudinal Study of Ageing (ELSA) dataset [30]. The data were obtained from the UK Data Service archive upon registration and can be found on the UK Data Service repository (https://ukdataservice.ac.uk/, accessed on 21 February 2023). Participants in ELSA were aged 50 and over and living in private households in England at the time of first recruitment. Beginning in 2000, they were assessed every 2 years using self-report questionnaires and face-to-face interviews (for more details, see https://www.elsa-project.ac.uk/, accessed on 21 February 2023). Ethical approval was obtained from the National Research Ethics Service, and informed consent was obtained from all participants. An additional study-specific ethical approval for secondary data analyses was provided by the College of Health, Medicine and Life Sciences Research Ethics Committee at Brunel University of London (ref: 36181-MHR-May/2022-39353-2).

The data for the participants selected in this study are from Wave 8 (2016–2017) because this was the first time self-reported sexual orientation was collected. Participants were included if they met the following criteria: (1) aged 50 or older at Wave 8; (2) non-institutionalized at both time points; (3) provided self-reported sexual orientation; (4) availability of data on socio-demographic characteristics (i.e., age, level of education, sex) at Wave 8; (5) availability of cognitive scores at both Waves 8 and 9; (6) no diagnosis of either dementia or another neurodegenerative disease potentially leading to cognitive decline (i.e., AD, Parkinson’s disease, motor neuron disease and multiple sclerosis) at both time points. In total, 8445 respondents were available at Wave 8 and 8736 at Wave 9. After the selection criteria were accounted for, the final sample included 4272 participants.

### 2.2. Sexual Orientation

Self-reported sexual orientation was assessed at Wave 8 with the question “Which of the following options best describes how you think about yourself?” [31]. If the participants answered “heterosexual or straight”, they were included in the HOA group; if they answered either “gay or lesbian”, “bisexual” or “other”, they were included in the SMOA group. Participants who either indicated that they preferred not to say their sexual orientation or did not complete this question were excluded. The final sample included 97.8% HOAs and 2.2% SMOAs. Given the small sample size of the SMOA group, sexual orientation sub-groups could not be analysed due to limited statistical power.

### 2.3. Psycho-Social Risk Factors for Cognitive Decline

Depressive symptoms were quantified by means of the Center for Epidemiologic Studies Depression Scale (CES-D) [32]. The CES-D includes 8 items to self-assess mood-related complaints in the previous week. Items were scored dichotomously (yes = 1; no = 0). The sum of all self-reported complaints resulted in a total score between 0 and 8 [33].Loneliness was self-assessed by means of the UCLA 3-Item Loneliness Scale [34]. This scale comprises three questions: “How often do you feel lack of companionship?”, “How often do you feel left out?” and “How often do you feel isolated from others?” Possible answers are (A) hardly ever, (B) some of the time and (C) often, corresponding to scores from 1 to 3, leading to a total loneliness score between 3 and 9.Participants’ marital status was coded as a binary variable to distinguish participants in a relationship from those who were not. At Wave 8, legal marital status was assessed by providing a series of response options: (A) single, that is, never married and never registered in a same-sex civil partnership; (B) married, first and only marriage or a civil partner in a registered same-sex civil partnership; (C) remarried, second or later marriage; (D) separated, but still legally married or (spontaneous only) in a same-sex civil partnership; (E) divorced or (spontaneous only) formerly in a same-sex civil partnership; (F) widowed or (spontaneous only) a surviving civil partner from a same-sex civil partnership. Due to small sample sizes across the sub-categories, marital status was operationalized as a binary variable to distinguish the participants currently in a relationship (Options B and C) from those not in a relationship for any reason (Options A, D, E and F).Socio-economic status (SES) was operationalised as quintiles of the total net non-pension wealth at the benefit unit level, in line with previous research [35]. Total net non-pension wealth has been extensively used as an index of SES because it captures current socio-economic circumstances and the wealth accumulated over the life course by older adults, associated with health outcomes [36,37].

Depression and loneliness symptoms have been selected because they are considered to be modifiable risk factors for cognitive decline [38] and may be particularly relevant to SMOAs. Indeed, depression prevalence [39] and loneliness severity [40] were found to be higher in SMOAs than in HOAs. Similarly, SMOAs are more likely not to have a romantic partner and to be living alone than HOAs [29], thus suggesting marital/relational status may be an important contributor to the health of SMOAs. Finally, SES was included since it is a social determinant of health inequalities in sexual minority groups [41]. In fact, non-heterosexual people have been found to earn less than heterosexual people (especially less than heterosexual men) [42].

### 2.4. Cognitive Outcome Measures

#### 2.4.1. Subjective Cognitive Decline

SCD for memory (SCD-mem) and for global cognition (SCD-cog) were assessed at Wave 8 using the following questions: for SCD-mem, “Compared to two years ago, would you say your memory is …?” and, for SCD-cog, “Compared to two years ago, would you say your other mental abilities are …?”. The possible answers were: (A) better now, (B) about the same and (C) worse now than they were then. Answers A and B were clustered together to distinguish participants who remained cognitively stable from those who reported a decline, independently of whether they thought their cognition was the same as or better than 2 years before. As a result, SCD-mem and SCD-cog were operationalized as binary variables (yes/no).

#### 2.4.2. Objectively Assessed Cognitive Decline

Objective cognitive decline was investigated for episodic and semantic memory by combining, for each domain, multiple tests collected at both waves. Episodic memory deficits are the most common symptom in AD; however, a decline in this cognitive domain is normally observed as a result of the normal ageing process [43]. In contrast, semantic memory is more stable in healthy older adults, and a decline, especially in lexico-semantic retrieval processes, as probed by the semantic fluency task, has been consistently documented as an early neuropsychological marker of AD [44,45]. Three measures of episodic memory were used.

A four-point measure (total score 0–4) assessing orientation to time (respondents were asked to provide (1) the day of the month, (2) the month, (3) the year and (4) the day of the week) [46].The word-list learning test [47] was used to assess how many words out of a list of 10 were recalled by participants both immediately after the reading (immediate recall, score 0–10) and after a delay of 5 min, during which the participants were asked other survey questions (delayed recall, score 0–10).The two measures assessing semantic memory were the following:Semantic fluency—animals [48] was used to assess the fluency of retrieval from the stored conceptual knowledge of a specific category of items. In particular, the participants were asked to report as many animals as they could in 60 s.Object and people naming was used to assess the participants’ conceptual knowledge by asking them to provide an answer to 5 definitions of either objects or people (score 0–5). This measure was adapted by combining and integrating items included in the telephone interview for cognitive status [49].

To assess objective cognitive change between the two waves, two reliable change indices (RCI), one for each memory domain, were computed, accounting for practice effects by using the standard error of the difference as a correction factor to account for variability in the scores at follow-up [50]. First, the raw test scores were summed up for each memory domain to obtain two global episodic and semantic memory performance scores. Then, the two RCIs, one for episodic memory (RCI-epi) and one for semantic memory (RCI-sem), were calculated by using the following formula:$$RCI=\frac{\left(T2-T1\right)-(M2-M1)}{\surd (SEM1+SEM2)}$$

Time 1 (T1) and time 2 (T2) represent the individual global memory score (either episodic or semantic) at Wave 8 and Wave 9, respectively; the average global memory scores of the ELSA sample at Time 1 (M1) and Time 2 (M2) (either episodic or semantic), and the associated standard error of scores of the ELSA sample at Time 1 (SEM1) and Time 2 (SEM2) are also included. In line with a previous similar study [18], the mean practice effect (M2 − M1) was calculated using all ELSA participants with cognitive data collected in the first 2 consecutive waves available, irrespectively of any missing data. Baseline and follow-up data were from Waves 1 and 2 for episodic memory, and from Waves 7 and 8 for semantic memory. As a result, the RCIs represent z-scores of changes in cognitive performance corrected for practice effects, where positive values indicate an improvement and negative values a decrease in performance from Wave 8 to Wave 9.

### 2.5. Statistical Analysis

Demographic, psycho-social and health profiles were compared between SMOAs and HOAs, after inspection of the results of the Shapiro–Wilk test, by using the Mann–Whitney U test for continuous variables that were not normally distributed. The chi-square (χ^2^) test was used for categorical variables.

Aim 1 was addressed by using 2 generalised linear models (GLMs), one for SCD-mem and one for SCD-cog, including: (I) self-reported sexual orientation and socio-demographic covariates (i.e., age, sex and educational qualification), (II) each risk factor (i.e., CES-D, marital status, loneliness and SES) and (III) the interaction between sexual orientation and each risk factor. The participants’ educational qualifications were operationalized as an ordinal variable, in line with a previous publication on ELSA data [16], by using a 6-level categorization based on the system used in England, Wales and Northern Ireland (https://www.gov.uk/what-different-qualification-levels-mean/list-of-qualification-levels, accessed on 2 March 2023).

Aim 2 was addressed by using 2 GLMS, one for RCI-epi and one for RCI-sem, including: (I) self-reported sexual orientation and socio-demographic covariates (i.e., age, sex and educational qualification), (II) each risk factor (i.e., CES-D, marital status, loneliness and SES) and (III) the interaction between sexual orientation and each risk factor.

Considering the large difference in the sizes of the two samples, a sensitivity analysis was carried out by selecting 2 samples of 92 HOAs and 92 SMOAs, individually matched for age, education and sex, and repeating all statistical models without the demographic variables (i.e., age, education and sex). Statistical analyses were carried out using IBM SPSS Statistics version 26 (IBM, Chicago, IL, USA). Since 4 GLMs were investigated, a Bonferroni’s corrected significance threshold of *p* = 0.0125 (i.e., *p* = 0.05/4) was used.

## 3. Results

### 3.1. Between-Group Comparison of Socio-Demographic Characteristics and Risk Factors

SMOAs and HOAs had similar demographic profiles, except for sex proportions, with the SMOA group including more males than the HOA group (Table 1). SMOAs were also less likely to be in a relationship than HOAs, while no between-group differences were found for all of the other risk factors.

### 3.2. Impact of Risk Factors on SCD in SMOAs and HOAs (Aim 1)

No differences in SCD-mem or SCD-cog prevalence were found between the SMOA and HOA groups (Table 2). Among the socio-demographic characteristics, older age was associated with higher odds of SCD-mem and SCD-cog. A weak effect of education was also detected: participants with the highest level of educational attainment, i.e., university degree or equivalent, had higher odds of reporting SCD-cog than participants with no certifications.

Among the risk factors, only more severe depressive symptoms were associated with higher risk of SCD-mem and SCD-cog (Table 2), yet similarly in both groups (Figure 1). No significant interaction effects were found between sexual orientation and any of the risk factors.

### 3.3. Impact of Risk Factors on Objective Cognitive Changes in SMOAs and HOAs (Aim 2)

In line with the findings on SCD, no differences in objective cognitive decline over time were found between SMOAs and HOAs (Table 3). Age was negatively associated with changes in both episodic and semantic memory. Additionally, depressive symptoms and SES (only in the comparison between the third with the first quintile; Appendix A, Figure A1) were negatively associated with changes in semantic memory only.

A significant interaction effect was found between depression and sexual orientation: depressive symptoms were associated with semantic memory decline only in the SMOA group (Figure 2).

### 3.4. Sensitivity Analysis in Matched Samples

When the two groups matched for age, education and sex were compared with one another, SMOAs were significantly more likely not to be in a relationship and reported higher levels of loneliness (Appendix B, Table A1).

More severe depression was associated with a higher risk of SCD-mem, but not of SCD-cog (Appendix B, Table A2), as well as with semantic memory decline (Appendix B, Table A3). Sexual orientation was not associated with any of the outcome measures, and no significant interaction effects between sexual orientation and any risk factors were found.

## 4. Discussion

This study found that the risk profiles of the two groups of older adults were relatively similar, with a difference in that SMOAs were less likely to be in a relationship. When the two groups were matched for demographic characteristics, this finding was replicated. Despite this difference, marital status was not associated with either SCD or cognitive decline in either group. This finding is in contrast with previous evidence suggesting a protective effect of being in a relationship on cognitive decline in both HOAs and SMOAs [5,16]. It is possible that the operationalization of marital status as a binary variable might have prevented the detection of more specific effects. In fact, Liu et al. [25] found a higher risk of cognitive impairment in people in SSRs who were cohabiting rather than being legally married. The lack of such fine-grained information in the ELSA dataset prevented the identification of people in a romantic yet not legally recognized relationship. Indeed, living alone may be a contributing factor to SCD risk [51].

As for subjectively reported cognitive problems, the severity of depressive symptoms was the only risk factor associated with a higher likelihood of SCD for both memory and global cognition equally in SMOAs and HOAs. This effect was confirmed in the sensitivity analysis for SCD-mem only and it provided further support to prior knowledge that people with low mood, independently of their sexual orientation, may be more likely to report cognitive problems subjectively in the absence of any detectable deficits [5,11,52,53]. Although this association may be interpreted as the consequence of greater worries about their cognition in people with more severe depressive symptoms, higher CES-D scores were also negatively associated with longitudinal changes in semantic memory. Moreover, the negative impact on semantic memory decline was significantly stronger in the SMOA than in the HOA group. Although not replicated in the sensitivity analysis, perhaps due to similarly mild depressive scores across the two groups, this is the first source of evidence that low mood may exert a greater negative impact on cognitive performance in SMOAs than in HOAs. Therefore, it is possible that the association between depression and cognitive health detected in this study may have emerged because, among all the risk factors considered, mood alterations represent the most sensitive index to the minority stress [54] experienced by SMOAs.

Differently from previous investigations that highlighted significant increases in cognitive decline risk in older people reporting higher levels of loneliness (even after controlling for depressive symptoms) [55,56], this study found no impact of loneliness on cognitive health. It must be noted that a weak association between loneliness and RCI-sem was observed, although not surviving Bonferroni’s correction. Contrary to expectations, this association was found to be positive, i.e., higher loneliness was associated with greater RCI-sem scores and was stronger in SMOAs. In this study, which is the first investigation of the potential impact of loneliness on the cognitive health of SMOAs, 50% of participants reported no loneliness, with a score of 3 (i.e., the lowest on a range from 3 to 9) on the UCLA three-item loneliness scale, and 75% of the sample had a score between 3 and 5. This suggests a very mild loneliness profile that may have prevented the detection of meaningful effects of loneliness on cognitive outcome measures. Future investigations on SMOA samples with more deeply characterised social profiles will be needed to elucidate the role played by loneliness, but also by social contact/isolation, on cognition.

Regarding SES, no associations were found with SCD, a finding similar to that of a study by Zullo et al. [53], and not in line with recent evidence of significant associations between higher SES and lower SCD risk [5,57] in SGM populations. A significant effect of SES, however, was found on changes in semantic memory. This was evident only in the comparison between the least wealthy people (lowest SES Quintile #1) with those in the middle quintile (SES Quintile #3) of the sample. Those in the lowest quintile showed higher RCI-sem scores (indicative of improvements over time), while those in the middle quintile showed a decline in semantic memory. This effect is against expectations of a protective influence of higher SES on cognitive health, such as reduced risk of cognitive decline [58] and of dementia [59] in ELSA. Consistently, a recent meta-analysis has also shown an increased risk of MCI and all-cause dementia in people with low SES [60]. The unexpected effect of SES that emerged in this study was not replicated in the sensitivity analysis and, therefore, it cannot be ruled out that this finding may be due to a selection bias and unbalanced sample sizes in the group comparison.

Net wealth quintiles are commonly used to investigate SES; in this study, they were applied to minimize potential issues related to the highly skewed distributions of net wealth observed in both participant groups. However, it must be mentioned that the distribution of participants across SES quintiles is in line with that reported by a previous investigation that included a larger sample of ELSA participants (*n* = 11566) [35]. Further investigations into different SES indices (e.g., income and availability of material resources) grounded in theoretical frameworks (e.g., material deprivation [42,61]) should be carried out to assess its potential protective influence (or lack of) on the health of SMOAs. To date, in fact, no SES gold standard exists, thus limiting any definite interpretation of these findings. Only a previous study found significantly worse cognitive performance in people over 50 in SSRs, compared with people in DSRs, exclusively among participants with high SES [21]. However, such a difference was only found cross-sectionally, while trajectories of decline over time were similar for the two groups, irrespective of SES.

Finally, among the demographic characteristics, older age was negatively associated with all outcome measures, i.e., higher SCD risk and lower RCI scores, in line with well-established age-related cognitive decline. Having the highest level of educational attainment, instead, was associated with greater odds of SCD-cog. This finding may appear counterintuitive because higher education is a well-known predictor of better cognitive health outcomes [38]. Since education was associated only with SCD risk but not with the severity of cognitive changes, it could be argued that highly educated participants, compared with people with no qualifications, may have had a higher degree of awareness of any cognitive changes. People with higher education might become aware of cognitive difficulties because they are in cognitively high-demanding jobs more often than people with lower education. Indeed, education was operationalized as an ordinal variable to capture not only the amount of time spent in education but also the complexity/advancement level of the qualifications achieved: the higher the level, the greater the likelihood of an individual being in occupations characterized by high cognitive demands. However, this and any other interpretations of this finding can only be speculative, given the lack of any measures of metacognitive abilities and of the biomarkers of neurodegenerative diseases in the ELSA database.

This study is not without limitations. First, the sample size of the SMOA group was relatively small, especially compared with that of the HOA group. However, significant findings were largely confirmed in the sensitivity analysis carried out by one-to-one matching HOAs to SMOAs for all demographic characteristics, particularly the negative association between the severity of depressive symptoms and different indices of cognitive decline. Second, different sexual orientation sub-groups were not investigated, potentially missing more specific effects. Indeed, bisexual people may be more likely to experience worse health, as may people not willing to disclose their sexual orientation [62]. However, the small sample sizes prevented any statistically meaningful sub-group analyses. Third, data on other risk factors that are potentially more relevant to SMOAs (e.g., discrimination due to sexual orientation) were lacking at Wave 8. Although everyday discrimination was assessed at Wave 5, data on such variables were missing for most SMOAs included in this study. Fourth, since data on clinical diagnosis were self-reported and given the lack of biomarkers for neurodegenerative diseases in the ELSA database, it cannot be completely ruled out that some participants might have had underlying pathologies consistent with neurodegenerative conditions, although at a pre-clinical stage. Fifth, it must be noted that the time elapsed between assessments was very short (about 2 years), thus limiting the interpretation of the long-term trajectory of cognitive changes in the participants included, who all entered this observational longitudinal study as cognitively unimpaired older adults and mostly expected to show minor variations in performance [63].

## 5. Conclusions

Overall, this study found an impact of depressive symptoms on self-reported and objective measures of cognitive decline. Low mood was associated with a decline in semantic memory more strongly in SMOAs than in HOAs. Clinicians should prioritise complementary assessments of the mental health of SMOAs presenting to memory clinic services with subjective cognitive complaints to ensure a comprehensive characterisation, informed by affirming clinical practices [64], of the potential factors influencing cognitive performance.

Future studies should clarify more in depth what factors may affect or protect from cognitive decline, not only in SMOAs but also in gender-diverse people, by collecting prospectively relevant and theory-informed variables. Applications of advanced methods, such as mediation analysis, are also encouraged to gain useful insights and advance the understanding of the possible interactions between different factors associated with cognitive decline among SGM older adult groups [65]. Moreover, clinical trials designed to improve the mental health of older sexual and gender minority groups could offer a preferential window to study the relationship between mental and cognitive health and to devise preventive strategies against cognitive deterioration that are potentially useful to the general aging population.

## Figures and Tables

**Figure 1 brainsci-15-00090-f001:**
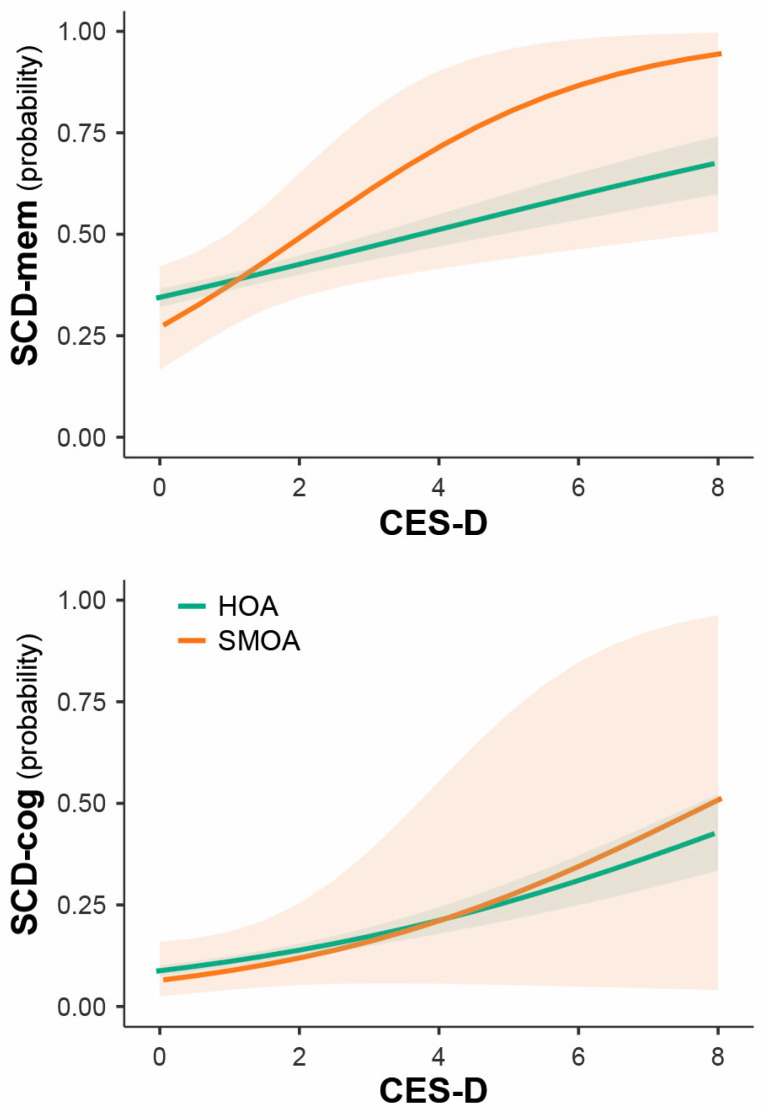
Impact of depressive symptoms’ severity on SCD risk in SMOAs and HOAs.

**Figure 2 brainsci-15-00090-f002:**
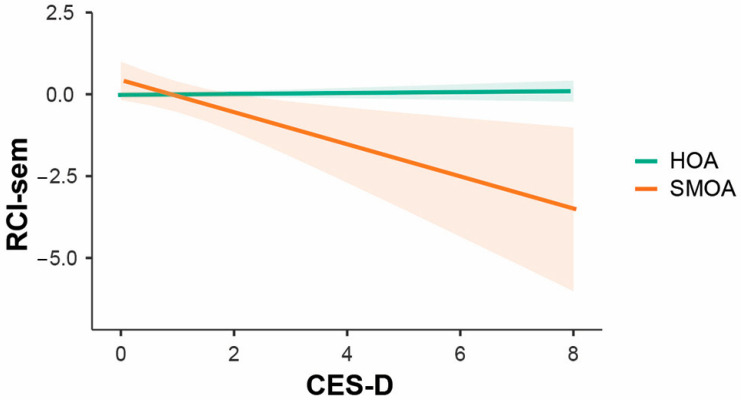
Differential impact of depressive symptoms’ severity on RCI-sem in SMOAs and HOAs.

**Table 1 brainsci-15-00090-t001:** Comparison of the socio-demographic characteristics and risk factors between the two samples.

Variable	HOA (*n* = 4180)	SMOA (*n* = 92)	Test	*p*
Age (years) ^a^	67.0 (11)	68.0 (12)	0.33 ^b^	0.739
Sex (M/F) ^c^	**43.8/56.2**	**54.3/45.7**	**4.08 ^d^**	**0.043**
Education ^c^				
No qualifications	8.9	12.0	10.97 ^d^	0.089
Level 1	11.4	12.0		
Level 2	34.1	22.8		
Level 3	12.3	16.3		
Level 4	14.8	12.0		
Level 5	2.9	1.1		
Level 6	15.6	23.9		
CES-D ^a^	0.0 (2)	1.0 (1)	−0.66 ^b^	0.511
Loneliness ^a^	3.0 (2)	3.0 (3)	−1.86 ^b^	0.063
Marital status (R/NR) ^c^	**69.1/30.9**	**46.7/53.3**	**20.96 ^d^**	**<0.001**
SES ^c^			3.15 ^d^	0.533
Quintile 1	19.9	24.7		
Quintile 2	20.1	14.6		
Quintile 3	20.0	21.3		
Quintile 4	20.1	16.9		
Quintile 5	19.9	22.5		

^a^ Median (interquartile range). ^b^ Mann–Whitney U test (standardized test statistic). ^c^ Percentages. ^d^ Chi-square test. In bold: variables that were significantly different between groups. CES-D, Center for Epidemiologic Studies Depression Scale; F, females; HOA, heterosexual older adults; M, males; NR, not in a relationship; R, in a relationship; SES, socio-economic status; SMOA, sexual minority older adults.

**Table 2 brainsci-15-00090-t002:** Results of the generalised linear models to test the effects of risk factors on SCD for memory and global cognition.

	SCD-mem		SCD-cog	
Variable	OR (95% CI)	*p*	OR (95% CI)	*p*
**Demographics**				
SO (SMOA)	1.70 (0.40, 7.27)	0.468	0.35 (0.03, 2.87)	0.356
Age	**1.03 (1.02, 1.04)**	**<0.001**	**1.03 (1.02, 1.04)**	**<0.001**
Sex (F)	1.07 (0.94, 1.22)	0.320	0.91 (0.75, 1.11)	0.366
Education (ref: no qualifications)				
Level 1	1.08 (0.80, 1.44)	0.618	1.22 (0.79, 1.88)	0.374
Level 2	1.09 (0.85, 1.40)	0.477	1.44 (1.01, 2.09)	0.047
Level 3	1.07 (0.80, 1.43)	0.663	1.50 (0.98, 2.30)	0.062
Level 4	1.37 (1.03, 1.82)	0.030	1.28 (0.84, 1.96)	0.261
Level 5	1.24 (0.80, 1.91)	0.325	1.14 (0.56, 2.19)	0.694
Level 6	1.35 (1.02, 1.79)	0.039	**1.78 (1.19, 2.71)**	**0.006**
**Risk factors**				
CES-D	**1.38 (1.14, 1.72)**	**0.002**	**1.35 (1.08, 1.73)**	**0.011**
Loneliness	0.95 (0.78, 1.13)	0.587	1.16 (0.90, 1.48)	0.246
Marital status (R)	0.75 (0.44, 1.23)	0.262	1.55 (0.73, 3.41)	0.259
SES (ref: Q1)				
Quintile 2	1.37 (0.60, 3.20)	0.458	1.20 (0.43, 3.41)	0.713
Quintile 3	1.70 (0.80, 3.80)	0.180	0.56 (0.11, 1.95)	0.394
Quintile 4	1.23 (0.55, 2.76)	0.601	0.81 (0.24, 2.51)	0.720
Quintile 5	1.29 (0.61, 2.82)	0.510	0.91 (0.28, 2.83)	0.873
**Interaction effects**				
CES-D × SO	1.35 (0.93, 2.09)	0.137	1.08 (0.70, 1.78)	0.727
Loneliness × SO	0.80 (0.55, 1.15)	0.251	1.20 (0.72, 1.96)	0.461
Marital status × SO	0.38 (0.13, 1.02)	0.060	1.38 (0.31, 6.72)	0.674
SES × SO				
(Q2 − Q1) × SO	2.01 (0.38, 10.99)	0.410	1.84 (0.24, 14.71)	0.551
(Q3 − Q1) × SO	2.72 (0.60, 13.63)	0.205	0.41 (0.02, 4.97)	0.513
(Q4 − Q1) × SO	1.50 (0.30, 7.47)	0.613	1.03 (0.09, 9.74)	0.980
(Q5 − Q1) × SO	2.46 (0.55, 11.73)	0.244	1.42 (0.14, 13.52)	0.754

CES-D, Center for Epidemiologic Studies Depression Scale; F, females; Q, quintile; R, in a relationship; SCD-mem/cog, subjective cognitive decline for memory/global cognition; SES, socio-economic status; SMOA, sexual minority older adults; SO, sexual orientation. Reference categories are heterosexual older adults (for sexual orientation), males (for sex), no qualifications (for education), not being in a relationship (for marital status), and quintile 1 (for socio-economic status). Significant effects that survived Bonferroni’s correction (*p* < 0.0125) are highlighted in bold.

**Table 3 brainsci-15-00090-t003:** Results of the generalised linear models to test the effects of risk factors on objective cognitive decline in episodic and semantic memory.

	RCI-epi		RCI-sem	
Variable	β (95% CI)	*p*	β (95% CI)	*p*
**Demographics**				
SO (SMOA)	0.20 (−0.76, 1.16)	0.678	−1.23 (−2.57, 0.10)	0.070
Age	**−0.01 (−0.02, −0.01)**	**<0.001**	**−0.02 (−0.03, −0.02)**	**<0.001**
Sex (F)	0.05 (−0.04, 0.015)	0.283	0.03 (−0.10, 0.16)	0.655
Education (ref: no qualifications)				
Level 1	−0.16 (−0.37, 0.04)	0.123	0.14 (−0.15, 0.42)	0.347
Level 2	−0.12 (−0.30, 0.05)	0.168	0.10 (−0.15, 0.34)	0.441
Level 3	−0.13 (−0.33, 0.08)	0.230	0.15 (−0.14, 0.44)	0.306
Level 4	−0.11 (−0.31, 0.10)	0.305	0.19 (−0.09, 0.47)	0.184
Level 5	−0.16 (−0.47, 0.15)	0.306	0.08 (−0.35, 0.51)	0.721
Level 6	−0.17 (−0.37, 0.03)	0.093	0.05 (−0.23, 0.33)	0.744
**Risk factors**				
CES-D	0.03 (−0.10, 0.15)	0.680	**−0.24 (−0.41, −0.06)**	**0.008**
Loneliness	−0.05 (−0.17, 0.07)	0.455	0.19 (0.02, 0.36)	0.026
Marital status (R)	0.21 (−0.32, 0.36)	0.905	−0.11 (−0.59, 0.36)	0.643
SES (ref: Q1)				
Quintile 2	0.09 (−0.46, 0.64)	0.754	−0.44 (−1.21, 0.32)	0.258
Quintile 3	0.14 (−0.38, 0.65)	0.600	**−0.95 (−1.67, −0.24)**	**0.009**
Quintile 4	0.10 (−0.42, 0.63)	0.703	−0.72 (−1.45, 0.01)	0.053
Quintile 5	−0.35 (−0.86, 0.16)	0.177	−0.56 (−1.27, 0.15)	0.123
**Interaction effects**				
CES-D × SO	0.03 (−0.23, 0.28)	0.832	**−0.50 (−0.86, −0.15)**	**0.005**
Loneliness × SO	−0.09 (−0.33, 0.15)	0.482	0.42 (0.09, −0.75)	0.014
Marital status × SO	−0.04 (−0.73, 0.64)	0.897	−0.40 (−1.35, 0.55)	0.413
SES × SO				
(Q2 − Q1) × SO	−0.12 (−1.22, 0.99)	0.834	−1.08 (−2.61, 0.46)	0.169
(Q3 − Q1) × SO	0.14 (−0.88, 1.17)	0.783	−1.74 (−3.17, −0.31)	0.017
(Q4 − Q1) × SO	0.18 (−0.86, 1.23)	0.738	−1.35 (−2.80, 0.11)	0.070
(Q5 − Q1) × SO	−0.93 (−1.94, 0.09)	0.075	−1.29 (−2.71, 0.13)	0.074

CES-D, Center for Epidemiologic Studies Depression Scale; F, females; Q, quintile; R, in a relationship; SCD-mem/cog, subjective cognitive decline for memory/global cognition; SES, socio-economic status; SMOA, sexual minority older adults; SO, sexual orientation. Reference categories are heterosexual older adults (for sexual orientation), males (for sex), no qualifications (for education), not being in a relationship (for marital status), and quintile 1 (for socio-economic status). Significant effects that survived Bonferroni’s correction (*p* < 0.0125) are highlighted in bold.

## Data Availability

The data used in the study were obtained from the UK Data Service archive upon registration and can be found on the UK Data Service repository (https://ukdataservice.ac.uk/).

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
