# Peer review of "Differential Impact of Risk Factors for Cognitive Decline in Heterosexual and Sexual Minority Older Adults in England"

_brainsci, 2025, doi:10.3390/brainsci15010090_

Round 1
Reviewer 1 Report
Comments and Suggestions for Authors
It is an interesting topic.
However, there are many aspects that need to be improved.
Lines 21-22: ,,Keywords: cognitive decline; memory; sexual orientation; minority; mental health; marital status; loneliness; socio-economic status”
I think there are many keywords.
Lines 116-120: ,, Ethical approval was obtained from the National Research Ethics Service and informed consent was obtained from all participants. An additional study-specific ethical approval for secondary data analysis was provided by the College of Health, Medicine and Life Sciences Research Ethics Committee at Brunel University London (ref: 36181-MHR-May/2022- 39353-2).”
If ethical approval was obtained, why was additional ethical approval needed? Can you comment?
In the discussion chapter you mentioned various aspects of the study. You talk about some of them in the subchapter limitations of the study. The limited number of subjects in a study does not allow for an appropriate statistical analysis. On the other hand, the lack of separate investigation of the different subgroups of sexual orientation did not allow for the issuance of data following the research. Another aspect you mention is the lack of biomarkers for the diagnosis of neurodegenerative diseases, possibly pre-existing in the research subjects.
I think clinical trials and other reviews are needed, because the topic is very interesting.
I think the conclusions can also be improved.
My comments are only intended to make the paper better. Good luck!
Reviewer 2 Report
Comments and Suggestions for Authors
The authors of this manuscript transversely compared the impact of psycho-social risk factors over cognitive performance in two healthy groups of heterosexual versus OVERALL diverse sexual orientations. The manuscript's objective of studying the potential effects of differences over memory and global cognition, aside from cognitive decline in episodic and semantic memory, is relevant and of great interest since its findings could provide rock-solid support for later studies markedly focused on clinical outcomes. Of course, I agree with the authors that future studies should explore and inquire about specific and individualized sexual subgroup orientations.
The manuscript text is straightforward and has acceptable parsimony. Although the paper shows an extensive and comprehensive methodology, it is still easy to read and follow specific descriptions.
The authors successfully connected the study´s findings with their general introduction, so, unsurprisingly, they took an interesting approach to their main question. At this stage, the future proposals they are suggesting have strong and enough evidence to be crystallized.
I have just a few recommendations:
1. In Table 1, I suggest grouping the education levels into two categories and repeating the analysis. (because they are using the Chi-Squared test).
2. In Table 1, why not use the total net non-pension wealth figure instead of quintile percentages and look for differences? Please explain and justify.
3. These two observations should be appropriately explained in the discussion section.
Reviewer 3 Report
Comments and Suggestions for Authors
This study investigates cognitive decline in sexual minority older adults (SMOAs), a highly underrepresented population in neurological research. This focus on psycho-social risk factors, particularly depressive symptoms, as differential drivers of cognitive decline is novel and addresses a significant gap in the literature. The manuscript represents a valuable contribution to the field. It addresses an important and timely topic with a well-designed study and offers actionable insights for both researchers and clinicians. Howvere, I will give my comments for each section in this manuscript.
Regarding the title is accurately reflecting the focus of the study, highlighting the comparison of cognitive decline between heterosexual and sexual minority older adults and emphasizing the investigation of risk factors. However, it lacks specificity regarding the study settings. The study utilized secondary data analysis based on participants from the English Longitudinal Study of Ageing (ELSA). The data were collected in England, United Kingdom, providing a clear geographical context and source of the study's population. So, please make sure to including this detail in the title that could enhance the specificity and clarity of the study's geographical context.
The introduction effectively contextualizes the research by addressing gaps in cognitive health studies among sexual minority older adults. It provides a clear rationale for the study, supported by relevant and up-to-date references. I am so glad this Introduction provides sufficient background for researchers outside the immediate specialty to understand the study. It explains the relevance of cognitive decline research among sexual minority older adults (SMOAs) and situates the study within the broader context of minority health disparities. However, the concept of "minority stress" could be briefly explained for broader accessibility.
Also, you need to clarify how the minority stress model informs the study design and its significance for cognitive health research.
The methods are described in detail, including sample selection, measures of cognitive decline, and the statistical analyses employed. The research design, utilizing reliable change indices and general linear models, is appropriate for assessing both subjective and objective cognitive decline. It accounts for key demographic and psycho-social variables, enhancing the study's rigor. Howvere, please clarify whether participants were evenly distributed across geographic regions or socioeconomic backgrounds within the ELSA dataset to enhance understanding of representativeness.
The results are clearly organized and presented, with tables and figures that summarize the findings effectively. Iam so glad aouthors include use of sensitivity analyses which adds robustness to the presentation.
The Discussion aligns well with the aims of the study. Some parts of the Discussion reintroduce general background on minority stress and health disparities. These could be condensed or moved to the Introduction for better focus on the results in the Discussion.
The authors acknowledge several key limitations of this study very well. For example, the small sample size of sexual minority older adults (SMOAs) reduces statistical power and limits the ability to detect nuanced differences between groups. Additionally, they mention the reliance on self-reported sexual orientation and cognitive measures introduces the potential for bias. The study's generalizability is further constrained by the predominantly UK-based and older sample, which may not fully represent broader populations is adressed very well. Other limitations could be considered include the possible underrepresentation of SMOAs with severe cognitive decline, as such individuals may have been less likely to participate in the English Longitudinal Study of Ageing (ELSA).
The conclusion is written very well and supported by the good data presention.
Overall, this manuscript is well-organized and effectively conveys a clear and focused message.
